# Therapeutic Targeting Steroid Resistant Pro-Inflammatory NK and NKT-Like Cells in Chronic Inflammatory Lung Disease

**DOI:** 10.3390/ijms20061511

**Published:** 2019-03-26

**Authors:** Greg Hodge, Sandra Hodge

**Affiliations:** 1Lung Research Unit, Department of Thoracic Medicine, Royal Adelaide Hospital, Adelaide 5001, Australia; sandra.hodge@adelaide.edu.au; 2Department of Medicine, University of Adelaide, Adelaide 5001, Australia

**Keywords:** steroid resistant NK and NKT-like cells, chronic inflammatory lung disease, COPD, BOS, IFNγ and TNFα, CD28null

## Abstract

The innate immune system drives the initiation of inflammation and progression to chronic inflammation in two important chronic inflammatory lung diseases involving the small airways, chronic obstructive pulmonary disease (COPD) and bronchiolitis obliterans syndrome (BOS), following lung transplantation. Recently natural killer T cell like (NKT-like) cells, which represent a bridge between the innate and adaptive immune response as well as the innate natural killer cell (NK) cells, have been shown to be important cells in these two chronic lung diseases. Importantly these cells have been shown to be resistant to commonly used anti-inflammatory drugs such as glucocorticoids and as such their inflammatory nature has been difficult to suppress. Mechanisms leading to steroid resistance in both diseases has recently been shown. Glucocorticoids switch off inflammatory genes by first entering the cell and binding to glucocorticoid receptors (GCRs). The steroid-GCR complex must then be chaperoned into the nucleus via several heat shock proteins, where they engage histone deacetylase 2 to switch off pro-inflammatory gene transcription. Many of these mechanisms are altered in NK and NKT-like cells in COPD and BOS requiring novel intervention using combinations of currently available drugs. Evidence will be presented to show how these drugs can overcome these mechanisms of drug resistance ex vivo advising novel therapeutic strategies for the treatment these two important chronic inflammatory lung diseases.

## 1. Introduction

Resistance to corticosteroids is an important barrier for the effective treatment of chronic obstructive pulmonary disease (COPD), and no currently available treatments slow disease progression, systemic inflammation, or associated increase in co-morbidity [1]. Similarly, immunosuppression therapy fails to prevent chronic graft failure in many patients following a lung transplant, with incidences of bronchiolitis obliterans syndrome (BOS) being 50% after five years [2]. Although the etiologies of both diseases are different, there are many striking similarities between the functional changes in several lymphocyte subsets that may play an important role in these two common steroid resistant diseases of the small airways. Although much of the research to date has focused on cells of the innate immune system, such as macrophages and neutrophils, that are involved in the inflammatory process of both these conditions, recent evidence has identified several lymphocyte subsets that may be critical in progressing these diseases. While T cells, particularly cluster of differentiation (CD)8+ T cells, have been identified as key players in the inflammatory responses of both diseases [3,4,5,6,7,8,9], other lymphocyte subsets such as the natural killer T cell like cells (NKT-like) and the innate natural killer cell (NK) cells are increasingly being recognized as playing important roles in the progression of these diseases and hence may be important therapeutic targets. A previous report by the current authors describes functional changes in steroid resistant CD8+CD28nullNKT-like pro-inflammatory cytotoxic cells in COPD [10]. However, the current review describes previously unreported changes in NK cells in this disease and in BOS, following lung transplantation. Current therapeutics fail to suppress the pro-inflammatory nature of these two important lymphocyte subsets in these chronic inflammatory lung diseases. NKT-like cells comprise an important yet uniquely small subset of lymphocytes that express features of both T and NK cells (Figure 1 and Figure 2) that represent a bridge between innate and adaptive immunity. These cells are distinct from invariant NKT cells (iNKT), a unique subset of T cells reactive to CD1d that recognize glycolipid antigens rather than peptides [11]. Conflicting reports of NKT-like cells in the blood of patients with COPD have shown them to be decreased [12], unchanged [13], or increased [14], although limitations of some studies were due to the lack of further immunotyping into CD4+ and CD8+ subsets. There have been reports of increases in NKT-like cells and NK cells in the bronchoalveolar lavage (BAL) and induced sputum of COPD patients and importantly these have been shown to be cytotoxic to autologous lung epithelial cells [12,13,15]. NKT-like cells and NK cells have also been reported to be increased in number as well as being a major source of pro-inflammatory cytokines and the cytotoxic molecules granzyme b and perforin following lung transplant [13]. Notably, these cells were increased in the small airways in patients with BOS when compared with stable transplant patients and healthy aged-matched controls [16].

There have been reports of increased numbers of CD8+ T cells producing interferon gamma (IFNγ) and tumor necrosis factor alpha (TNFα) pro-inflammatory cytokines in the peripheral blood in the BAL and lungs from COPD ex-smoker and current-smoker COPD groups compared with healthy smokers and control groups, indicating the systemic pro-inflammatory nature of this disease [3] These changes were recorded regardless of whether patients were receiving inhaled corticosteroids, thus demonstrating a lack of effectiveness of current therapies at reducing these pro-inflammatory cytokines. Unfortunately, further immunophenotyping of NKT-like and NK cells was not performed, and this would be an important area of further research. Steroid resistance of these cells in vitro was identified by failure of 0.1–1 µM dexamethasone to suppress the production of IFNγ by CD8+ T cells. Again, subtyping of NKT-like and NK cells was not performed and as such this would be a valuable area to further investigate [17].

Steroid resistance was shown to be similarly increased in patients with BOS compared with stable transplant patients and in stable patients compared with healthy aged-matched control subjects as assessed by the decreased inhibitory effect of 1 µM prednisolone on IFNγ and TNFα production by CD8+ T cells [18]. Unfortunately, NKT-like and NK cells were not assessed.

The main aim of this review is to identify functional changes in steroid resistant NKT-like and NK cells in patients with COPD and BOS with the aim of improving therapeutic strategies to overcome steroid resistance in two major chronic inflammatory lung diseases that respond inadequately to current immunosuppression protocols.

## 2. Loss of CD28 on Steroid Resistant Senescent NKT-Like Lymphocytes in COPD and BOS

The loss of co-stimulatory molecule CD28 on T and NKT-like cells following persistent chronic antigenic stimulation in patients with COPD has been reported [19]. The origin of peptide antigens responsible for initiating this loss of CD28 is speculative but has been suggested to include microbial peptides as bacterial colonization of the airways is frequent in COPD [20], tobacco related peptides, elastin peptides, and other autoantigens [20]. Loss of CD28 from CD4+ and CD8+ T cells from patients with BOS [21] was also increased when compared with stable patients and was shown to correlate with increased granzyme b, IFNγ, and TNFα production as well as increased steroid resistance. Further immunophenotyping with NKT-like markers was not performed but given the recent report of increased cytotoxic, pro-inflammatory cytokines in NK and NKT-like cells in the small airways in patients with BOS, such a study would be worthwhile [16].

## 3. Increased Transmembrane Pump, P-Glycoprotein 1, In NK and NKT-Like Cells

For therapeutic drugs to be effective at reducing the pro-inflammatory/cytotoxic potential of steroid resistant lymphocytes, they must first overcome P-glycoprotein 1 (Pgp1), a transmembrane efflux pump which has been well characterized in drug resistant cancer cells [22]. Increased numbers of Pgp1+ peripheral blood NKT-like and NK cells co-expressing IFNγ, TNFα, and granzyme b have been reported in COPD patients compared with healthy aged-matched controls [23] (Figure 3). The authors showed an inverse correlation between Pgp1 expression and calcium-AM uptake, a dye sensitive to Pgp1 efflux [23]. Further investigations by the same authors revealed increased Pgp1 expression by both CD8+CD28nullNKT-like and CD8+CD28+NKT-like subsets (unpublished). Importantly, treatment with a known Pgp1 inhibitor, cyclosporine A and standard dose corticosteroid, prednisolone (1 µM) resulted in synergistic inhibition of both IFNγ and TNFα production by steroid resistant NKT-like and NK cells from COPD patients [23] (Figure 3). Furthermore, the dose of cyclosporine A required to inhibit Pgp1 was 2.5 ng/mL, which is approximately 25 times less than the dose used to treat transplant rejection so not considered to be associated with any unwanted side effects. There has been a very recent report on Pgp1 expression in lymphocyte subsets in patients with BOS [18]. Pgp1 expression was increased in CD8+Pgp+ T cells and correlated with IFNγ/TNFα expression along with the BOS grade [18]. However, Pgp1 expression was decreased in NKT-like and NK lymphocyte subsets in patients with BOS compared with heathy aged-matched control subjects suggesting alternate mechanisms of steroid resistance in these patients [18].

Glucocorticoids enter cells by overcoming membrane drug efflux pump P-glycoprotein-1 (Pgp1) and binding to the glucocorticoid receptor (GCR) in the cytoplasm. GCR must be bound to the molecular chaperones heat shock proteins (Hsp)70 and Hsp90 to acquire a high-affinity steroid binding conformation, and trafficked to the nucleus where engagement of histone deacetylases (HDACs), particularly HDAC2, results in the reduction of pro-inflammatory gene activation.

In COPD and BOS patients compared with age-matched healthy control subjects the percentage of steroid resistant CD28nullCD8+NKT-like cells and NK cells are increased.

A1. Pgp1+ NKT-like cells are increased in COPD (but not BOS), reducing intracellular levels of glucocorticoid (GC). Expression of GCR (B1), HDAC2 (C1) and Hsp90 (D1) are decreased in CD28nullCD8+NKT-like cells and NK cells (with no change in Hsp70) in COPD and BOS patients reducing steroid effectiveness.

Possible therapeutic targeting to overcome steroid resistance in CD28nullCD8+NKT-like cells and NK cells in COPD and BOS:A2. Pgp1 is synergistically decreased in the presence of 2.5 ng/mL cyclosporine A (CsA) and 1 µM prednisolone in the blood of COPD patients ex vivo.B2. GCR expression is increased in the presence of 2.5 ng/mL cyclosporine A (CsA) and 1 µM prednisolone in the blood of COPD and BOS patients ex vivo.C2. HDAC2 expression is increased in the presence of 5 mg/mL theophylline, 2.5 ng/mL CsA and 1 µM prednisolone in the blood of COPD and BOS patients ex vivo.D2. Hsp90 expression is increased in the presence of 2.5 ng/mL CsA and 1 µM prednisolone in the blood of COPD and BOS patients ex vivo.E. Therapeutic targeting results in decreased IFNγ and TNFα pro-inflammatory cytokine expression in CD28nullCD8+NKT-like cells and NK cells in the blood of COPD and BOS patients ex vivo.

## 4. Loss of Glucocorticoid Receptor in NKT-like and NK Cells

Following entry into the cell, glucocorticoids must bind to the GCR before transportation into the cell nucleus. Decreased expression of GCR has been reported in CD28nullNKT-like cells in patients with COPD compared with healthy aged-matched controls [19] (Figure 3). Loss of CD28 was associated with an increased percentage of NKT-like cells producing both IFNγ and TNFα and also a loss of GCR [19] (Figure 1). There was a significant correlation between IFNγ and TNFα production and GCR expression by CD28nullNKT-like cells in both patients with COPD and controls [24]. However, the percentage of CD28nullNKT-like cells was significantly greater in patients with COPD compared with the control subjects [19]. Although GCR expression was not reported in NK cells in this study, further analysis showed that GCR expression was also reduced in NK cells from COPD patients compared with aged-matched control subjects (Figure 2). Furthermore, there was also a significant correlation between IFNγ and TNFα production and GCR expression by NK cells in patients with COPD (unpublished data). There was a significant negative correlation between disease severity, as shown with forced expiratory volume in 1 s (FEV_1_), and the percentage of GCR negative CD28nullNKT-like cells [8] and NK cells (unpublished data) in patients with COPD. These data suggests that alternate treatment options to glucocorticoids are required to suppress pro-inflammatory cytokines in these patients. Reduced GCR has been reported in CD8+NKT-like and NK cells from stable lung transplant patients and patients with BOS compared with controls [18]. Another study of patients with interstitial lung disease showed decreased expression of GCR mRNA in lung tissue from steroid resistant compared with steroid sensitive patients [24]. Hence, the use of GCR activators may be useful in treatment of steroid resistance in COPD and BOS and possibly other chronic inflammatory lung diseases. Recent unpublished findings by Hodge et al. showed that expression of GCR in NK and NKT-like cells from COPD and lung transplant patients was upregulated in the presence of 1 µM prednisolone and 2.5 ng/mL cyclosporine A, revealing another previously unknown mechanism of action of these drugs (Figure 3).

## 5. Decreased Histone Deacetylase 2 in NKT-Like and NK Cells

HDAC2 is required by corticosteroids to switch of activated inflammatory genes. HDAC2 has been shown to be reduced in peripheral blood pro-inflammatory CD8+CD28nullNKT-like cells in patients with COPD [25] (Figure 1). HDAC2 expression was also shown to negatively correlate with the percentage of CD8+CD28nullNKT-like cells producing IFNγ and TNFα [26]. Although HDAC2 expression was not assessed in NK cells from these patients, further analysis showed that their HDAC2 expression was also decreased in NK cells (Figure 3) and negatively correlated with IFNγ and TNFα production by these cells (unpublished data). Addition of low dose theophylline (5mg/L), a HDAC2 activator, synergistically upregulated HDAC2 in CD8+CD28nullNKT-like cells in the presence of 1µM prednisolone and also cyclosporine A (2.5 ng/mL) and resulted in decreased production of IFNγ and TNFα by these cells [25]. Similar findings were noted for NK cells (unpublished data). HDAC2 has also been reported to be decreased in CD8+NKT-like cells following lung transplant [26] and importantly in steroid resistant CD8+NKT-like and NK cells in the small airways in patients with BOS compared with stable patients and controls [27] which correlated with disease severity (FEV_1_). Addition of low dose theophylline and 1 µM prednisolone resulted in the synergistic upregulation of HDAC2 in CD8+NKT-like cells [27] (Figure 3) and NK cells (unpublished data) also suggesting that therapeutic upregulation of HDAC2 in these lymphocytes may reduce steroid resistance and inflammation caused by these cells and improve graft survival. Importantly, identifying CD8+NKT-like and NK cells in small airway brushings using flow cytometry [27] overcomes the limitations associated with inadequate biopsy sampling techniques [28].

## 6. Decreased Heat Shock Protein in NKT-Like and NK Cells

The glucocorticoid-GCR complex must be bound to molecular chaperones Hsp 70 and Hsp90 to acquire the high affinity steroid binding conformation to traffic to the cell nucleus [29]. A recent study investigated the expression of both Hsp70/90 in CD8+NKT-like cells from the peripheral blood of patients with COPD [30]. While expression of Hsp70 was unaltered, expression of Hsp90 and GCR was reduced from CD8+CD28nullNKT-like cells (Figure 1) and correlated with the cytotoxic/pro-inflammatory nature of these cells and also lung function (FEV_1_) of these patients [30]. Addition of low dose cyclosporine A (2.5ng/mL), know to selectively bind to Hsp90 and not Hsp70, resulted in reduced production of IFNγ and TNFα by CD8+CD28nullNKT-like cells in combination with 1µM prednisolone [30] (Figure 3). Analysis of NK cells was not performed in this study but further analysis showed decreased Hsp90 in NK cells (Figure 2), increased expression of Hsp90 in the presence of these drugs and reduced production of these pro-inflammatory cytokines by these cells (Figure 3). Given the similarities in functional changes in NK and NKT-like cells between patients with COPD and BOS, a study examining Hsp90 in lung transplant patients may prove worthwhile.

## 7. Future Therapy for Chronic Inflammatory Lung Disease

Lymphocyte senescence and glucocorticoid resistance have been described in a subset of difficult to treat asthma patients, another chronic inflammatory disease involving the small airways [31], and several other inflammatory conditions such as cardiovascular disease [32], autoimmune disease [33], arthritis [34], inflammatory bowel disease, aging [35], and aging associated with COPD [36].

Several of these are also co-morbid conditions associated with COPD and hence may also be associated with increased pro-inflammatory NKT-like cells and NK cells. The report of increased cytotoxicity to airway epithelial cells by both NK and NKT-like cells in the airways of patients with COPD points directly to the damage that these cells are capable of causing [13]. Targeting these functional changes associated with steroid resistance and inflammation, by decreasing Pgp expression and upregulating GCR, HDAC2 and Hsp90 in NKT-like and NK cells, may improve patient morbidity and reduce the comorbid conditions associated with COPD. Targeting these functional changes in these cells at an early stage when NKT-like and NK cell numbers are increasing in the small airways is likely to be important to prevent apoptosis of lung cells, associated fibrosis and eventual falling lung function. Hence timing of such interventional therapeutics will prove critical [37].

The report of increased expression of CD8+ T cell granzyme b, IFNγ, and TNFα in the peripheral blood of lung transplant patients many months before a fall in lung function and subsequent diagnosis of BOS points to possibly the best biomarkers to date to predict impending chronic allograft rejection [38,39]. NK markers were not performed in these studies but given the numerous subsequent studies reported by Hodge et al., it is highly likely that NK and NKT-like cells will be the cytotoxic/pro-inflammatory culprits causing epithelial cell destruction in these patients. These simple laboratory tests are available worldwide and argue for serial monitoring of all lung transplant patients so that interventional therapeutics can be commenced immediately to prevent these cells causing lung epithelial cell death and associated fibrosis [40] with loss of lung function. Similarly, reports of increased expression of CD8+ T cell IFNγ and TNFα in peripheral blood, BAL and bronchial brushings in patients with COPD and the differential expression of these cytokines by these cells between trachea and bronchi also argues for further studies including NK markers to determine if NK and NKT-like cells are also causing cell death in the lungs of COPD patients and if similar biomarkers may predict which smokers will progress to COPD [3,7].

This review identifies the effectiveness of several drugs such as low dose cyclosporine to target multiple mechanisms associated with steroid resistance and may also prove invaluable in the treatment of several other common inflammatory diseases. Such early interventional studies as those proposed in this review may prove critical to prevent cell death in the lungs and subsequent progression of these currently untreatable debilitating diseases and reduce associated health care costs.

## Figures and Tables

**Figure 1 ijms-20-01511-f001:**
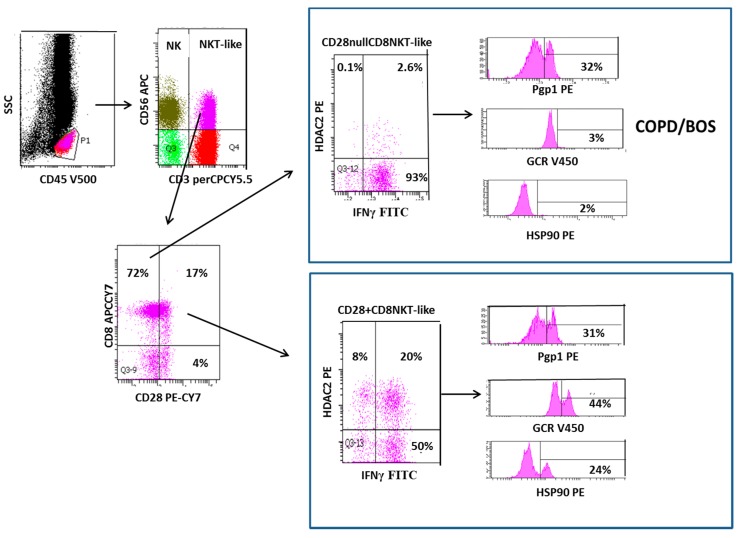
Flow cytometric gating technique used to identify cluster of differentiation (CD)28nullCD8+natural killer T cell like (NKT-like) cells. Identification of lymphocytes as CD45+ low side scatter (SSC) events; Identification of NKT-like cells as CD3+CD56+ events; Identification of CD8+ NKT-like cells using CD8 APC-CY7 staining; Identification of CD28nullCD8+NKT-like cells using CD28 PE-CY7 staining; expression of interferon gamma (IFNγ) and histone deacetylase 2 (HDAC2) in CD28nullCD8+NKT-like cells and CD28+CD8+NKT-like cells; Expression of P-glycoprotein 1 (Pgp1), glucocorticoid receptor (GCR) and heat shock protein (Hsp90) expression in CD28nullCD8+NKT-like cells and CD28+CD8+NKT-like cells. Note: The percentage of CD28nullCD8+NKT-like cells are increased in patients with chronic obstructive pulmonary disease (COPD) and bronchiolitis obliterans syndrome (BOS). CD28nullCD8+NKT-like cells express reduced HDAC2, GCR and Hsp90 but increased IFNγ compared with CD28+CD8+NKT-like cells (Pgp1 unchanged). This figure was adapted from [10].

**Figure 2 ijms-20-01511-f002:**
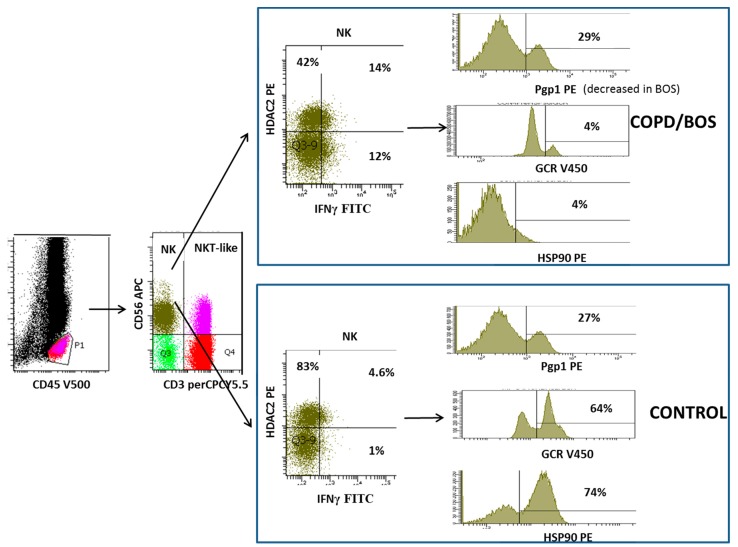
Flow cytometric gating technique used to identify natural killer cell (NK) cells. Identification of lymphocytes as CD45+ low SSC events; Identification of NK cells as CD3-CD56+ events. Expression of IFNγ and HDAC2 in NK cells from COPD and BOS patients compared with controls. Expression of Pgp1, GCR and Hsp90 expression in NK cells from patients with COPD and BOS compared with controls. Note: NK cells express reduced HDAC2, GCR and Hsp90 but increased IFNγ in patients with COPD and BOS compared with controls. Pgp1 expression was unchanged in patients with COPD but decreased in patients with BOS compared with control subjects (not shown) suggesting alternate mechanisms of steroid resistance in these patients.

**Figure 3 ijms-20-01511-f003:**
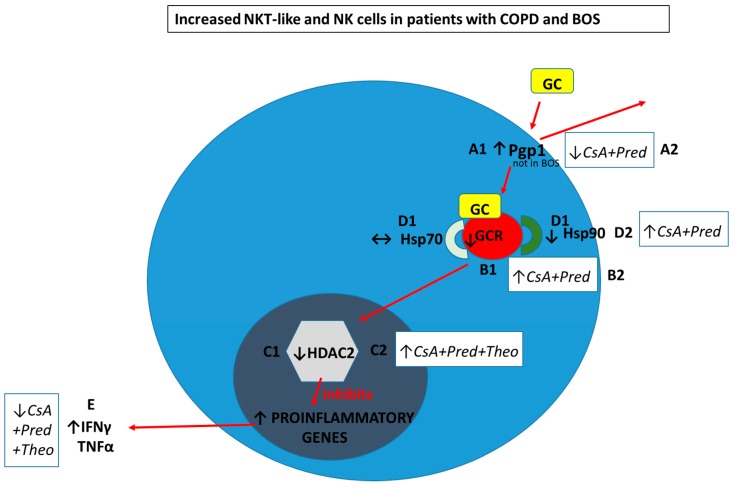
Schematic diagram summarizing reported findings in peripheral blood CD28nullCD8+NKT-like cells and NK cells in COPD and BOS patients compared with aged-matched control subjects. The percentage of CD28nullCD8+NKT-like cells and NK cells are increased in COPD and BOS patients compared with aged-matched control subjects. This figure was adapted from [10].

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
