# Peer review of "Therapeutic Targeting Steroid Resistant Pro-Inflammatory NK and NKT-Like Cells in Chronic Inflammatory Lung Disease"

_ijms, 2019, doi:10.3390/ijms20061511_

Round 1

Reviewer 1 Report

The review by Hodge G. refers to corticosteroids resistance of COPD and BOS, due to altered expression of proteins that are part of the anti-inflammatory machinery in NK and NKT-like cells.

The issue is of interest but it would be preferable to give a wider scientific overview of the topic, by adding more literature, if any, in support. It might be rather limiting that twenty out of forthy-one references are self-citations. The title might be misleading, since the authors did not discuss functional changes other than the anti-inflammatory activity. Moreover, they assess that one of the aim of this review is to present the currently available drugs that could overcome the corticosteroids resistance. Thus, I would suggest to reconsider the title. Figure 3 is not so clear, capital letters and numbers render the picture rather confused. It could be useful to summarize the data in a table.

Author Response

Comment:

The issue is of interest but it would be preferable to give a wider scientific overview of the topic, by adding more literature, if any, in support. It might be rather limiting that twenty out of forthy-one references are self-citations.

Reply:

The authors have been studying functional changes in lymphocyte subsets in these two important chronic inflammatory lung diseases for the last 15 years and were the first to describe steroid resistance of NKT and particularly NK cells very recently in these diseases. As such, there is very little in the way of other reports describing the mechanisms associated with steroid resistance in these cells by researchers other than those already referenced in the manuscript.

Comment:

 The title might be misleading, since the authors did not discuss functional changes other than the anti-inflammatory activity. Moreover, they assess that one of the aim of this review is to present the currently available drugs that could overcome the corticosteroids resistance. Thus, I would suggest to reconsider the title.

Reply:

The title of the manuscript has been changed to “Therapeutic targeting steroid resistant pro-inflammatory NK and NKT-like cells in chronic inflammatory lung disease”

Comment:

Figure 3 is not so clear, capital letters and numbers render the picture rather confused. It could be useful to summarize the data in a table.

Reply:

Although the schematic is a little busy, the authors feel it was important to show where in NKT-like and NK cells these functional changes occur and the capitals to identify these changes and corresponding therapeutic drug combinations. Whereas a table may be even more confusing to non-cell biologists or scientists unfamiliar with these mechanisms.

Reviewer 2 Report

The authors wrote an interesting review that give an overview in this field underlining the importance to improve therapeutic strategies to overcome developed resistance to several drugs.

I suggest the authors revise references because about half of that are self-citations, so I think is important to differentiate their literature.

Author Response

Comment:

I suggest the authors revise references because about half of that are self-citations, so I think is important to differentiate their literature.

Reply (as for Reviewer 1):

The authors have been studying functional changes in lymphocyte subsets in these two important chronic inflammatory lung diseases for the last 15 years and were the first to describe steroid resistance of NKT and particularly NK cells very recently in these diseases. As such, there is very little in the way of other reports describing the mechanisms associated with steroid resistance in these cells by researchers other than those already referenced in the manuscript.

Reviewer 3 Report

The current manuscript by Greg Hodge and Sandra Hodge review the functional changes in steroid resistant NK and NKT like cells in chronic inflammatory lung disease including chronic obstructive pulmonary disease (COPD) and bronchiolitis obliterans syndrome (BOS). 

However, the work in this manuscript is similar to the one published by same authors in 2016 (Steroid Resistant CD8+CD28null NKT-Like Pro-inflammatory Cytotoxic Cells in Chronic Obstructive Pulmonary Disease, Frontiers in Immunology) except that bronchiolitis obliterans syndrome (BOS) is described in this manuscript along with COPD. Figure 1 (Flow gating strategy for NKT like cells) is exactly the same image and the overall concept of the cartoon is same as well. The sub topics correlate as well. 

In my opinion, this review does not contribute much to what is already published in fact by the same group previously. 

Author Response

Comment:

However, the work in this manuscript is similar to the one published by same authors in 2016 (Steroid Resistant CD8+CD28null NKT-Like Pro-inflammatory Cytotoxic Cells in Chronic Obstructive Pulmonary Disease, Frontiers in Immunology) except that bronchiolitis obliterans syndrome (BOS) is described in this manuscript along with COPD. Figure 1 (Flow gating strategy for NKT like cells) is exactly the same image and the overall concept of the cartoon is same as well. The sub topics correlate as well. 

In my opinion, this review does not contribute much to what is already published in fact by the same group previously. 

Reply:

Added to manuscript line 44 “A previous report by the current authors describes functional changes in steroid resistant CD8+CD28nullNKT-like pro-inflammatory cytotoxic cells in COPD [10]. However, the current review describes previously unreported changes in NK cells in this disease and importantly in these cell types in BOS following lung transplantation. Current therapeutics fail to suppress the pro-inflammatory nature of these two important lymphocyte subsets in these chronic inflammatory lung diseases.”

The flow gating strategy for NKT cells could have been referenced to this previous review (reference 10) but the authors felt it important to show the gating strategy and findings for both NK and NKT-like cells in this review. The authors agree the cartoon and subtopics are similar to the previous report but now describe and show functional changes for both NKT-like cells and previously unreported NK cells for both COPD and BOS. While we could have written a report on novel findings for NK cells, we felt it important to compare these two important lymphocyte subsets for these two steroid resistant chronic inflammatory lung diseases, especially showing similar therapeutic strategies in overcoming their pro-inflammatory nature.

Round 2

Reviewer 1 Report

Dear authors,

I appreciate the revised version of the present manuscript and the replies to my comments and suggestions.

Sincerely

Anna Caretti

Reviewer 3 Report

The authors justified previous comments. The only concern I still have is that the gating strategy figure 1 is a an exact one (exact meaning the same sample used for gating. Even the dots correlate) from the previously published work in Frontiers in Immunology. This might conflict the journal copyright. The authors have to clearly mention that it is adapted from the previous work and mention the article or use a gating strategy from a new sample. 

Also please check if IJMS allows using images from already published work.